# Dynamics of $\mathrm{CP}^{N-1}$ skyrmions

Seungho Lee,[1,2,∗] Hyojae Jeon,[1,†] and Jung Hoon Han[1,‡]

[1]*Department of Physics, Sungkyunkwan University, Suwon 16419, South Korea*
[2]*Institute of Basic Science, Sungkyunkwan University, Suwon 16419, South Korea*
(Dated: September 23, 2025)

We derive several exact results for the dynamics of $\mathrm{CP}^{N-1}$ skyrmions with arbitrary $N$. Fractonic continuity equation is shown to hold for arbitrary $\mathrm{CP}^{N-1}$ fluid implying the conservation of the topological charge and the dipole moment. Inclusion of the Gilbert damping modifies the continuity equation, resulting in the violation of the dipole moment conservation but not of the topological charge. Thiele's equation for the $\mathrm{CP}^{N-1}$ skyrmion follows from the modified continuity equation. The Girvin-MacDonald-Platzman (GMP) algebra in the long-wavelength limit is derived for arbitrary $\mathrm{CP}^{N-1}$ fluid. In the case of $\mathrm{CP}^2$ skyrmions, we identify two kinds of energetically stable skyrmions in which the quadrupolar moments or the ferromagnetic moments are dominant. In the latter case, one can associate a nonzero $\mathrm{CP}^1$ charge equal to half the $\mathrm{CP}^2$ skyrmion charge and argue that the topological Hall effect of electrons should exist due to their coupling to the ferromagnetic part of the $\mathrm{CP}^2$ texture.

## I. INTRODUCTION

Conventional treatments of magnetism and magnetic dynamics are framed in terms of O(3) classical spins or SU(2) quantum spins. Extensions of the scheme to include orbital and multipolar degrees of freedom in a spin-orbit-coupled system also exist. More recently, interesting developments have taken place to understand magnetic dynamics in multipolar systems [1–3] with quadrupolar and higher-order multipolar moments [4–8]. In a concurrent effort, extension of the theory of $\mathrm{CP}^1$ skyrmions to $\mathrm{CP}^{N-1}$ ($N > 2$) has taken place [9–15] with emphasis on the *structure* of $\mathrm{CP}^{N-1}$ skyrmions in multipolar magnets [10–13]. In this work, we explore various aspects of the *dynamics* of the $\mathrm{CP}^{N-1}$ fluid and of the isolated $\mathrm{CP}^{N-1}$ skyrmions. Along the way, several known results for $\mathrm{CP}^1$ fluids such as the fractonic conservation laws [16, 17] and topological Hall effect [18, 19] of skyrmions are extended to $\mathrm{CP}^{N-1}$.

Magnetic skyrmions have been discovered in a host of ferromagnets [20–23]. The skyrmions discovered to date are the $\mathrm{CP}^1$ skyrmions defined by the homotopy $\pi_2(\mathrm{CP}^1) = \mathbb{Z}$. In spin-1 magnets, on the other hand, the five magnetic quadrupolar moments as well as the three magnetic dipole moments together form an eight-component order parameter, represented by $\mathrm{CP}^2$ field. The possibility of $\mathrm{CP}^2$ skyrmion formation in spin-1 magnets has been scrutinized recently [10–12]. A suggestion to create $\mathrm{CP}^3$ skyrmions in a magnetic bilayer also exists [14]. It is becoming clear that $\mathrm{CP}^{N-1}$ theory should provide a unifying framework to address magnetic dynamics of multipolar magnets.

A strong analogy among the dynamics of electrons in the lowest Landau level, vortices in superfluids, and skyrmions in magnets has been noted for a long time [19, 24]. In a modern perspective, they can be viewed as manifestations of fractonic dynamics and hydrodynamics [17, 25] in diverse physical platforms. The Girvin-MacDonald-Platzman (GMP) algebra, originally conceived as a manifestation of the constrained dynamics of electrons in the lowest Landau level [26], has found close connection to fractonic field theory [27] and the theory of $\mathrm{CP}^1$ fluid [17].

In this paper, we show that the fractonic continuity equation for the $\mathrm{CP}^1$ fluid, originally derived in the early 90s [16], generalizes to $\mathrm{CP}^{N-1}$ with arbitrary $N > 2$. The violation of the fractonic conservation and the concomitant loss of dipole conservation, but not the topological charge conservation, in the presence of Gilbert damping is derived. The GMP algebra obeyed by the $\mathrm{CP}^1$ topological density [17] is shown to generalize to arbitrary $N$. These findings are reported in Sec. III. The question of topological Hall effect by mobile electrons coupled to the texture of $\mathrm{CP}^{N-1}$ skyrmion is tackled in Sec. IV, with focus on the $\mathrm{CP}^2$ skyrmion structure. Two types of $\mathrm{CP}^2$ skyrmions called quadrupolar skyrmions and ferromagnetic skyrmions are identified, with the latter type most likely to result in the topological Hall effect. Some mathematical backgrounds are summarized in Sec. II and in several appendices.

## II. MATHEMATICAL BACKGROUNDS

The generators $\lambda_a$ ($a = 1, \cdots, N^2 - 1$) of SU($N$) group obey the algebra [28]

$$\lambda_a \lambda_b = \frac{2}{N} \delta_{ab} \mathbf{1}_{N \times N} + (d_{abc} + i f_{abc}) \lambda_c, \qquad (2.1)$$

with the structure constants $f_{abc}$, the symmetric coefficients $d_{abc}$, and the normalization following from

$$f_{abc} = \frac{1}{4i} \mathrm{Tr}\left(\lambda_a [\lambda_b, \lambda_c]\right),$$

$$d_{abc} = \frac{1}{4} \mathrm{Tr}\left(\lambda_a \{\lambda_b, \lambda_c\}\right),$$

$$\delta_{ab} = \frac{1}{2} \mathrm{Tr}\left(\lambda_a \lambda_b\right). \qquad (2.2)$$

The $\mathrm{CP}^{N-1}$ field $\mathbf{z}$, in turn, is defined as $\mathbf{z} = U\mathbf{z}_0$ where $\mathbf{z}_0$ is a complex unit-norm $N$-component vector, and $U$ is an

---

∗ seungho6626@gmail.com
† wjsgywo980@gmail.com
‡ hanjemme@gmail.com

element of SU(N). A real-valued vector $n^a$ given by

$$n^a = \mathbf{z}^\dagger \lambda_a \mathbf{z} \qquad (2.3)$$

satisfies

$$n^a n^a = \frac{2(N-1)}{N}, \quad n^a = \frac{N}{2(N-2)} d_{abc} n^b n^c. \qquad (2.4)$$

Hereafter, repeated indices are implicitly summed over.

The topological three-current for $\text{CP}^{N-1}$ can be defined as

$$J_\mu = \frac{1}{8\pi} \varepsilon_{\mu\nu\lambda} f_{abc} n^a \partial_\nu n^b \partial_\lambda n^c = \frac{1}{2\pi} \varepsilon_{\mu\nu\lambda} \partial_\nu a_\lambda \qquad (2.5)$$

where $\mu = t, x, y$ and $a_\mu = -i\mathbf{z}^\dagger \partial_\mu \mathbf{z}$. The integral of the temporal component is the $\text{CP}^{N-1}$ topological charge

$$Q_{N-1} = \int_\mathbf{r} J_t \quad \left( \int_\mathbf{r} := \int d^2\mathbf{r} \right), \qquad (2.6)$$

which is an integer by virtue of the second homotopy $\pi_2(\text{CP}^{N-1}) = \mathbb{Z}$. Another useful identity is

$$\partial_\mu n^a \partial_\mu n^a = 4(D_\mu \mathbf{z})^\dagger (D_\mu \mathbf{z}), \qquad (2.7)$$

where $D_\mu = \partial_\mu - ia_\mu$ is the covariant derivative.

The dynamics of $\text{CP}^{N-1}$ magnets in the absence of damping is governed by the generalized Landau-Lifshitz (gLL) equation [1–3]

$$\partial_t n^a = 2f_{abc} \frac{\delta H}{\delta n^b} n^c \qquad (2.8)$$

which follows from the action

$$S = \frac{1}{2} \int d^2\mathbf{r} dt \int_0^1 du f_{abc} n^a \partial_u n^b \partial_t n^c - \int dt H, \qquad (2.9)$$

where $H$ is the Hamiltonian. The derivation of the gLL equation from the action $S$ is given in Appendix A. The LL equation for the O(3) magnet is recovered by substituting $f_{abc} = \varepsilon_{abc}$.

## III. FRACTONIC CONSERVATION LAW AND GMP ALGEBRA

### A. Conservation laws

In the absence of damping, the three-current $J_\mu$ in (2.5) satisfies the continuity equation $\partial_\mu J_\mu = 0$ for any $\text{CP}^{N-1}$ magnet purely by virtue of the algebra governing $n^a$; see Appendix B for derivation. As emphasized by Skyrme [29, 30], conservation laws that exist independent of the symmetry of the action is deemed *topological*. We can go further and re-write continuity equation for $\text{CP}^{N-1}$ fluid in the *fractonic* form $(i, j = x, y)$

$$\partial_t J_t + \partial_i \partial_j J_{ij} = 0, \qquad (3.1)$$

where a tensor current

$$J_{ij} = \frac{\varepsilon_{ik}}{2\pi} \sigma_{kj} \qquad (3.2)$$

is defined in terms of the stress tensor

$$\sigma_{kj} = \mathscr{H} \delta_{kj} - \frac{\partial \mathscr{H}}{\partial(\partial_j n^a)} \partial_k n^a \qquad (3.3)$$

for a given density $\mathscr{H}$ of the energy functional

$$H[\mathbf{n}] = \int_\mathbf{r} \mathscr{H}(\mathbf{n}, \partial_i \mathbf{n}). \qquad (3.4)$$

For derivation, see Appendix B.

Fractonic continuity equation (3.1) implies the conservation of the total dipole moment $\mathbf{D}$ in addition to the total charge:

$$\mathbf{D} = \int_\mathbf{r} \mathbf{r} J_t. \qquad (3.5)$$

Due to the dipole conservation, a single $\text{CP}^{N-1}$ skyrmion cannot move; instead only the skyrmion-antiskyrmion pair (a skyrmion dipole) with a fixed relative position can move freely. Two skyrmions of the same charge can rotate around each other while preserving their center of mass. The fractonic conservation law in (3.1) for $N = 2$ had been known since the early 90s [16] though the word 'fracton' did not exist at the time. A similar fractonic conservation law exists for the vortices in superfluids as well [25], highlighting the generality of the fractonic description for these states of matter. We have here shown that all $\text{CP}^{N-1}$ fluids obey the fractonic dynamics.

Including the Gilbert damping modifies the dynamics given by (2.8) and, accordingly, the fractonic conservation law. The gLL equation in the presence of Gilbert damping becomes

$$\partial_t n^a = 2f_{abc} \frac{\delta H}{\delta n^b} n^c - \alpha \frac{\delta H}{\delta n^a}$$
$$= 2f_{abc} \frac{\delta H}{\delta n^b} n^c - \alpha f_{abc} f_{cde} n^b n^d \frac{\delta H}{\delta n^e}, \qquad (3.6)$$

where $\alpha > 0$ represents damping [3, 31]. The continuity equation (3.1) is modified to

$$\partial_t J_t + \partial_i \partial_j J_{ij} = \frac{\alpha}{4\pi} \varepsilon_{ij} f_{abc} \partial_i \left[ \frac{\delta H}{\delta n^a} n^b \partial_j n^c \right].$$
$$= \frac{\alpha}{8\pi} \varepsilon_{ij} \partial_i [\partial_t n^c \partial_j n^c] + O(\alpha^2). \qquad (3.7)$$

For derivation of the modified continuity equation, see Appendix B. The second line follows from the first by invoking the gLL equation (2.8). Under the new continuity equation (3.7), the topological charge $Q_{N-1}$ (2.6) is still conserved but the dipole moment is not:

$$\frac{d}{dt} \mathbf{D} = -\frac{\alpha}{8\pi} \hat{x} \int_\mathbf{r} \partial_t \mathbf{n} \cdot \partial_y \mathbf{n} + \frac{\alpha}{8\pi} \hat{y} \int_\mathbf{r} \partial_t \mathbf{n} \cdot \partial_x \mathbf{n}. \qquad (3.8)$$

Further progress can be made for the *rigid* configuration, $\mathbf{n}(\mathbf{r}, t) = \mathbf{n}(\mathbf{r} - \mathbf{R}(t))$, such that $\partial_t \mathbf{n} = -(\dot{\mathbf{R}} \cdot \nabla)\mathbf{n}$. Inserting

this back into (3.8) and assuming $\int_{\mathbf{r}}(\partial_x\mathbf{n})^2 = \int_{\mathbf{r}}(\partial_y\mathbf{n})^2 = I$ and $\int_{\mathbf{r}} \partial_x\mathbf{n} \cdot \partial_y\mathbf{n} = 0$ as in a rotationally symmetric skyrmion gives an equation of motion for the dipole moment

$$\dot{\mathbf{D}} = \frac{\alpha I}{8\pi}\dot{\mathbf{R}} \times \hat{z} \implies \hat{z} \times \dot{\mathbf{D}} = \frac{\alpha I}{8\pi}\dot{\mathbf{R}}. \tag{3.9}$$

For a rigid skyrmion we can also write $\mathbf{D} = Q_{N-1}\mathbf{R}$, $\dot{\mathbf{D}} = Q_{N-1}\dot{\mathbf{R}}$, and recover Thiele's equation [32, 33]

$$Q_{N-1}\hat{z} \times \dot{\mathbf{R}} = \frac{\alpha I}{8\pi}\dot{\mathbf{R}} + \mathbf{F}_{\text{ext}} \tag{3.10}$$

in the absence of the external force $\mathbf{F}_{\text{ext}}$. For a single $CP^{N-1}$ skyrmion, even allowing for dissipation gives $\dot{\mathbf{R}} = 0$ unless there is an external force $\mathbf{F}_{\text{ext}}$ acting on it. This well-known result for $CP^1$ skyrmion is now shown to hold for arbitrary $CP^{N-1}$.

A more dramatic implication of the modified conservation law (3.7) is in regard to the creation or annihilation of the $CP^{N-1}$ skyrmion-antiskyrmion pair, which changes the dipole moment $\mathbf{D}$ and would not be possible under the strict fractonic conservation law of (3.1). The creation of skyrmion-antiskyrmion pair, or a skyrmion dipole, had been frequently seen in numerical simulations of LLG equation [34, 35], but its connection to the non-conservation of the dipole moment through the equation (3.8) had not been noted before.

### B. GMP algebra

A convenient way to capture the dynamics of the $CP^{N-1}$ order parameters $n^a$ is through the Poisson bracket $\partial_t n^a = \{n^a, H\}$ along with the fundamental relation

$$\{n^a(\mathbf{r}), n^b(\mathbf{r}')\} = 2f_{abc}n^c(\mathbf{r})\delta(\mathbf{r} - \mathbf{r}'), \tag{3.11}$$

which is the classical correspondence to the commutation relation of the SU(N) generators $[\lambda_a, \lambda_b] = 2if_{abc}\lambda_c$ [1]. With a given energy functional $H$, the gLL equation (2.8) follows from the Poisson bracket as

$$\frac{\partial n^a}{\partial t} = \{n^a, H\} = \int_{\mathbf{r}} \frac{\delta H}{\delta n^b}\{n^a, n^b\} = 2f_{abc}\frac{\delta H}{\delta n^b}n^c. \tag{3.12}$$

One can show, based on the fundamental relation (3.11), that

$$\{J_t(\mathbf{r}), \mathbf{n}(\mathbf{r}')\} = \frac{\varepsilon_{ij}}{2\pi}\partial_i\mathbf{n}\partial_j\delta(\mathbf{r} - \mathbf{r}') \tag{3.13}$$

and

$$\{J_t(\mathbf{r}), J_t(\mathbf{r}')\} = \frac{\varepsilon_{ij}}{2\pi}\partial_i J_t(\mathbf{r})\partial_j\delta(\mathbf{r} - \mathbf{r}') \tag{3.14}$$

for the topological charge density $J_t$—see Appendix C for derivation. This is the Girvin-MacDonald-Platzman (GMP) algebra, first derived for the density operators in the lowest Landau level to describe neutral excitations [26], for the topological density operators in the $CP^{N-1}$ fluid. A similar algebra had been constructed for $CP^1$ fluid [17], vortices in superfluids [36], and for the fractonic field theory [27]. The Poisson

bracket for the dipole moment follows from the GMP algebra (3.14) as

$$\{D_i, D_j\} = -\frac{\varepsilon_{ij}}{2\pi}Q_{N-1}. \tag{3.15}$$

To summarize, we have derived the fractonic conservation law for the topological three-current and its modification under damping, as well as the GMP algebra for the topological density, valid for arbitrary $CP^{N-1}$ fluid.

## IV. TOPOLOGICAL HALL EFFECT OF $CP^2$ SKYRMION

The theory of spin transfer torque is based on the strong Hund coupling of local spin texture $\mathbf{S}$ to the itinerant spin moment $\psi^\dagger\boldsymbol{\sigma}\psi$, both of which transform as O(3) vectors and can be coupled linearly. The two-component spinor $\psi = (\psi_\uparrow \ \psi_\downarrow)^T$ represents electrons of both spin orientations.

In $CP^{N-1}$ magnets, the $(N^2 - 1)$-component vector $n^a$ breaks down as three magnetic, five quadrupolar, seven octupolar components and so on totalling $N^2 - 1$ components. To couple higher-order local moments to mobile electrons, one must first form products of electron spins such as $(\psi^\dagger\sigma_a\psi)(\psi^\dagger\sigma_b\psi)\cdots$ with the indices $a, b = x, y, z$ arranged to form a matching multipole moment. Such interactions are inevitably of higher order in the Hund's interaction strength and are likely to grow weaker with the increasing order of multipoles. A reasonable assumption is that even in multipolar magnets, the main interaction channel between electron fluids and the local moments is through their dipolar Hund coupling. It is thus essential to inquire if a $CP^{N-1}$ skyrmion texture with $N > 2$ can also carry nonzero $CP^1$ topological charge. We address this question by constructing a quite general $CP^2$ skyrmion texture and examining its $CP^1$ charge.

A well-known "trick" for constructing a model $CP^1$ configuration is to use the Bogomol'nyi-Prasad-Sommerfeld inequality [37–40]. We first show that this trick applies for arbitrary $CP^{N-1}$ and use this result to construct $CP^2$ skyrmion configurations of arbitrary charge [15]. The inequality reads

$$(D_x\mathbf{z} + i\,\text{sgn}(Q_{N-1})D_y\mathbf{z})^\dagger(D_x\mathbf{z} + i\,\text{sgn}(Q_{N-1})D_y\mathbf{z}) \geq 0, \tag{4.1}$$

for $CP^{N-1}$ fields $\mathbf{z}$. The sign of the $CP^{N-1}$ charge $Q_{N-1}$ is explicitly included. Equivalently,

$$\sum_{j=x,y}|D_j\mathbf{z}|^2 \geq \frac{\text{sgn}(Q_{N-1})}{i}\left((D_x\mathbf{z})^\dagger D_y\mathbf{z} - (D_y\mathbf{z})^\dagger D_x\mathbf{z}\right). \tag{4.2}$$

Invoking the identity (2.7) and (2.5) and integrating over all space,

$$E \equiv \frac{1}{2}\int_{\mathbf{r}}\sum_{j=x,y}(\partial_j\mathbf{n})^2 \geq 4\pi|Q_{N-1}|. \tag{4.3}$$

For a given charge $Q_{N-1}$, the energy $E$ is minimized by achieving the equality $E = 4\pi|Q_{N-1}|$ when

$$D_x\mathbf{z} + i\,\text{sgn}(Q_{N-1})D_y\mathbf{z} = 0. \tag{4.4}$$

This equation is solved by

$$\mathbf{z} = \frac{\xi^{|n|}\mathbf{u} + r^{|n|}e^{in\theta}\mathbf{v}}{(\xi^{2|n|} + r^{2|n|})^{1/2}}, \tag{4.5}$$

where $n$ is the integer-valued $CP^{N-1}$ skyrmion charge $Q_{N-1}$, $\xi$ is the skyrmion radius, and $\mathbf{u}$ and $\mathbf{v}$ are two arbitrary $N$-dimensional complex unit vectors satisfying $\mathbf{u}^* \cdot \mathbf{v} = 0$ [4, 41, 42]. One can work out the corresponding gauge field

$$\mathbf{a} = \frac{nr^{2(|n|-1)}}{\xi^{2|n|} + r^{2|n|}}(-y, x) \tag{4.6}$$

and the covariant derivatives $D_\mu \mathbf{z} = (\partial_\mu - ia_\mu)\mathbf{z}$ to show that $D_x\mathbf{z} + i\,\mathrm{sgn}(n)D_y\mathbf{z} = 0$, and that the topological number is indeed $n$:

$$Q_{N-1} = \frac{2\xi^{2|n|}n|n|}{2\pi}\int \frac{r^{2|n|-1}}{(\xi^{2|n|} + r^{2|n|})^2}dr d\theta = n. \tag{4.7}$$

The $CP^{N-1}$ charge is $n$ regardless of the choice of $\mathbf{u}, \mathbf{v}$ vectors.

Specializing to $CP^2$, the skyrmion should have textures in both dipolar and quadrupolar sectors. The question we address now is, given the $CP^2$ charge $n$, would the texture also support nonzero $Q_1$? One must first keep in mind that, unlike the $CP^2$ charge, the $CP^1$ charge of the skyrmion solution (4.5) depends on the choice of $\mathbf{u}, \mathbf{v}$. Rather than making arbitrary choices for the two unit vectors, we use energy consideration to narrow down the scope of $(\mathbf{u}, \mathbf{v})$ vectors. For the energy functional, the previous energy $E = (1/2)\int\sum_j(\partial_j\mathbf{n})^2$ is supplemented with an anistropy:

$$E' = E + \kappa\int_{\mathbf{r}}\mathbf{S}^2 \tag{4.8}$$

where the magnetic dipole $\mathbf{S} = (S^x, S^y, S^z)$ can be worked out from $\mathbf{z}$ in (4.5) by $\mathbf{S} = \mathbf{z}^\dagger(\lambda_7, -\lambda_5, \lambda_2)\mathbf{z}$ (see Appendix D for details of $CP^2$ algebra). The task at hand is to find a pair of $(\mathbf{u}, \mathbf{v})$ that minimizes the functional

$$\kappa\int_{\mathbf{r}}\frac{\xi^{4|n|}4|\mathbf{u}_r\times\mathbf{u}_i|^2 + r^{4|n|}4|\mathbf{v}_r\times\mathbf{v}_i|^2 - 2\xi^{2|n|}r^{2|n|}(|\mathbf{u}\cdot\mathbf{v}|^2-1)}{(\xi^{2|n|} + r^{2|n|})^2}, \tag{4.9}$$

where $\mathbf{u}_r, \mathbf{u}_i, \mathbf{v}_r, \mathbf{v}_i$ are real vectors defined by $\mathbf{u} = \mathbf{u}_r + i\mathbf{u}_i$ and $\mathbf{v} = \mathbf{v}_r + i\mathbf{v}_i$.

For $\kappa > 0$, the energy is minimized by $(\mathbf{u}, \mathbf{v}) = (e^{i\alpha/2}, e^{i\beta/2})\hat{\mathbf{e}}$ for a pair of arbitrary phases $(\alpha, \beta)$ and a real-valued unit vector $\hat{\mathbf{e}}$ (see Appendix D for details of energy minimization). In this case, the local magnetization vector $\mathbf{S}$ vanishes identically and the skyrmion texture resides entirely in the quadrupolar sector. This type of $CP^2$ skyrmion will be called the *quadrupolar* $CP^2$ skyrmion. In hindsight, it is natural that the choice $\kappa > 0$ suppresses the magnetic part completely.

For $\kappa < 0$, as shown in Appendix E, the energy-minimizing configuration is parameterized by

$$\mathbf{u} = e^{i\varphi}\begin{pmatrix}\cos\alpha\sin\beta\\\sin\alpha\sin\beta\\\cos\beta\end{pmatrix}, \mathbf{v} = \frac{e^{-i\gamma}}{\sqrt{2}}\begin{pmatrix}\cos\alpha\cos\beta - i\sin\alpha\\\sin\alpha\cos\beta + i\cos\alpha\\-\sin\beta\end{pmatrix} \tag{4.10}$$

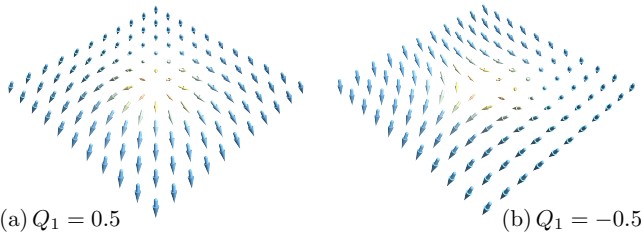

(a) $Q_1 = 0.5$         (b) $Q_1 = -0.5$

FIG. 1: Magnetic dipole texture $\mathbf{S}$ for $\alpha = \pi/4$, $\beta = \pi$, $\gamma = \pi/4$, $\varphi = 0$, and $\xi = 1$ in (4.10). (a) $n = 1$, (b) $n = -1$.

with arbitrary angles $(\alpha, \beta, \gamma, \varphi)$. The magnetic dipole and quadrupole profiles associated with such $(\mathbf{u}, \mathbf{v})$ can be worked out (see Appendix E), and the $CP^1$ skyrmion number following from its magnetic dipole texture is

$$Q_1 = \frac{1}{4\pi}\int_{\mathbf{r}}\frac{\mathbf{S}\cdot\partial_x\mathbf{S}\times\partial_y\mathbf{S}}{|\mathbf{S}|^3}$$
$$= \frac{1}{4\pi}\int\frac{2n|n|\xi^{2|n|}r^{|n|-1}}{(r^{2|n|} + 2\xi^{2|n|})^{3/2}}dr d\theta = \frac{n}{2}, \tag{4.11}$$

which is half the $CP^2$ skyrmion number.

The reason for the half-integer value of $Q_1$ above can be seen in Fig. 1 which shows the magnetic dipole configuration $\mathbf{S}$ for $n = \pm 1$. At first sight it is tempting to view these half-integer entities as merons when $n$ is an odd integer, but a closer examination reveals a more complicated picture. The size of the magnetic dipole moment is given by

$$\mathbf{S}^2 = \frac{r^{2|n|}(r^{2|n|} + 2\xi^{2|n|})}{(r^{2|n|} + \xi^{2|n|})^2} \tag{4.12}$$

independent of the angles parameterizing $(\mathbf{u}, \mathbf{v})$. The size of the magnetic dipole is depleted as $r^{|n|}$ near the origin and reaches 1 for $r/\xi \gtrsim 1$ where $\xi$ defines the core radius. Thus for $\kappa < 0$, the $CP^2$ skyrmion texture becomes mostly magnetic outside the the core and predominantly quadrupolar inside. We refer to such texture as *ferromagnetic* $CP^2$ skyrmion, to emphasize its largely magnetic character outside the core. A typical meron, on the other hand, would have the spin pointing along $\pm\hat{z}$ and winding around the equator at infinity, costing infinite magnetic energy. The origin of the half-integer value for $Q_1$ in (4.11) is attributed to the vanishing of the magnetic dipole moment at the origin.

The Hund coupling between local and itinerant moments and the theory of topological Hall effect derived from such coupling assumes that the magnitude $|\mathbf{S}|$ remains finite, which will be the case for the ferromagnetic $CP^2$ skyrmion in the region outside the core. We thus expect that ferromagnetic $CP^2$ skyrmions with nonzero $Q_1$ charge to exhibit topological Hall effect, despite the appearance of magnetic quadrupolar moments in the core.

## V. SUMMARY

We have identified a number of general properties of the $CP^{N-1}$ fluids including (i) the fractonic topological conserva-

tion law for the topological density, (ii) its modification under the Gilbert damping and consequent violation of the dipole moment conservation, and (iii) the validity of GMP algebra for the topological charge density and the dipole moment. All the properties hold for arbitrary $CP^{N-1}$ fluid independent of the details of the governing Hamiltonian.

The topological Hall effect of mobile electrons coupled to the ferromagnetic spin texture, well-known in the case of $CP^1$ skyrmions, is argued to hold for ferromagnetic $CP^2$ skyrmion but not for the quadrupolar $CP^2$ skyrmion. Though not investigated in detail, we conjecture that an arbitrary $CP^{N-1}$ skyrmion can be classified according to the dominance in ferromagnetic, quadrupolar, and higher multipole-moment sectors. Amongst these, the ferromagnetic $CP^{N-1}$ skyrmion with nonzero $CP^1$ charge will demonstrate topological Hall effect.

## ACKNOWLEDGMENTS

JHH was supported by the National Research Foundation of Korea (NRF) grant funded by the Korea government (MSIT) (Grant No. 2023R1A2C1002644 and No. RS-2024-00410027). We acknowledge delightful conversations with Dung Nguyen Xuan on the field-theoretic understanding of GMP algebra and its relation to fractonic hydrodynamics. SL thanks Seunghun Lee for discussions.

## Appendix A: Generalized Landau-Lifshitz-Gilbert equation

The dynamics of $CP^{N-1}$ skyrmions stems from the gLL, which is derived from the Poisson bracket with the Hamiltonian $\partial_t n^a = \{n^a, H\} = \int \delta H / \delta n^b \{n^a, n^b\} = 2f_{abc}(\delta H / \delta n^b)n^c$. In this section, we show the gLL can be formally introduced from the $CP^{N-1}$ Wess-Zumino action which is defined as

$$S_{WZ} = \int_{x^3} \int_0^1 du \frac{1}{2} f_{abc} n^a \partial_u n^b \partial_t n^c. \tag{A1}$$

where $\int_{x^3} = \int d^2\mathbf{r} dt$. The variation of $S_{WZ}$ gives $\delta S_{WZ} = \int_{x^3} f_{abc} \partial_t n^b n^c \delta n^a / 2$ and

$$\frac{\delta S_{WZ}}{\delta n^a} = \frac{1}{2} f_{abc} \partial_t n^b n^c. \tag{A2}$$

The equation of motion follows from the extremization of the total action $S = S_{WZ} - \int dt H$:

$$f_{abc} \partial_t n^b n^c = 2 \frac{\delta H}{\delta n^a}. \tag{A3}$$

One can multiply the left-hand side of Eq. (A3) by $f_{ade} n^d$ and invoke some identities

$$\partial_\mu n^a = \frac{N}{(N-2)} d_{abc} n^b \partial_\mu n^c$$

$$f_{abc} f_{ade} = \frac{2}{N} (\delta_{be} \delta_{cd} - \delta_{bd} \delta_{ce}) + d_{bea} d_{cda} - d_{bda} d_{cea} \tag{A4}$$

to get the simplified expression $f_{abc} f_{ade} \partial_t n^b n^c n^d = \partial_t n^e$. Multiplying the same factor to the right-hand side of Eq. (A3)

gives the gLL in the conventional form:

$$\partial_t n^e = 2 f_{ade} \frac{\delta H}{\delta n^a} n^d. \tag{A5}$$

The dynamics of $CP^{N-1}$ skyrmions in the presnece of dissipation can be captured by the generalized Landau-Lifshitz-Gilbert (gLLG) equation

$$\partial_t n^a = 2 f_{abc} \frac{\delta H}{\delta n^b} n^c - \alpha \frac{\delta H}{\delta n^a}$$
$$= 2 f_{abc} \frac{\delta H}{\delta n^b} n^c - \alpha f_{abc} f_{cde} n^b n^d \frac{\delta H}{\delta n^e} \tag{A6}$$

The equivalence of the damping terms in the first and second lines can be proved by employing the identity

$$\frac{\delta H}{\delta n^a} = \frac{N}{N-2} d_{abc} n^b \frac{\delta H}{\delta n^c}. \tag{A7}$$

Equation (A7) stems from the constraint [Eq. (2.4)]. From the constraint of the field, we directly obtain the constraint for tangent vectors

$$\dot{n}^a = \frac{N}{N-2} d_{abc} n^b \dot{n}^c, \tag{A8}$$

where $\dot{n}$ is a derivative with respect to an arbitrary parameter. Any tangent vector can be expressed as a linear combination of elements of a basis of the tangent space

$$\dot{n}^a = C^\alpha e^a_\alpha \tag{A9}$$

for some coefficients $C^\alpha$ and basis $\{\mathbf{e}_1, \mathbf{e}_2, \cdots, \mathbf{e}_{N^2-1}\}$. Then, from Eq. (A8), we have

$$C^\alpha e^a_\alpha = \frac{N}{N-2} d_{abc} n^b C^\alpha e^c_\alpha$$
$$e^a_\alpha = \frac{N}{N-2} d_{abc} n^b e^c_\alpha. \tag{A10}$$

Since the functional derivative also is a tangent vector of the order parameter $n^a$, it can be written as

$$\frac{\delta H}{\delta n^a} = D^\alpha e^a_\alpha \tag{A11}$$

for some coefficients $D^\alpha$ and, by substituting Eq. (A10) into Eq. (A11), yields Eq. (A7).

The change of the total energy governed by the gLLG is given by

$$\frac{dH}{dt} = \int_\mathbf{r} \frac{\delta H}{\delta n^a} \partial_t n^a$$
$$= \int_\mathbf{r} \left[ 2 f_{abc} \frac{\delta H}{\delta n^a} \frac{\delta H}{\delta n^b} n^c - \alpha \frac{\delta H}{\delta n^a} \frac{\delta H}{\delta n^a} \right]$$
$$= -\alpha \int_\mathbf{r} \left| \frac{\delta H}{\delta \mathbf{n}} \right|^2. \tag{A12}$$

Manifestly, the total energy decreases in the presence of the Gilbert damping.

## Appendix B: Topological current conservation of fractonic nature

Here we show $\partial_\mu J_\mu = 0$ for the topological three-current

$$J_\mu = \frac{\varepsilon_{\mu\nu\lambda}}{8\pi} f_{abc} n^a \partial_\nu n^b \partial_\lambda n^c . \tag{B1}$$

From the definition $n^a = \mathbf{z}^\dagger \lambda_a \mathbf{z}$ with $\mathbf{z} = U\mathbf{z}_0$, derivatives of the field $n^a$ can be expressed as

$$\partial_\mu n^a = \mathbf{z}^\dagger [U\partial_\mu U^\dagger, \lambda_a] \mathbf{z} = \frac{i}{2} \omega_\mu^b \mathbf{z}^\dagger [\lambda_b, \lambda_a] \mathbf{z} = f_{abc} \omega_\mu^b \mathbf{z}^\dagger \lambda_c \mathbf{z} = f_{abc} \omega_\mu^b n^c , \tag{B2}$$

by using $U\partial_\mu U^\dagger = i\omega_\mu^b \lambda_b / 2$ for some $\omega_\mu^b$ [43]. The divergence of $J_\mu$ is written as

$$\partial_\mu J_\mu = \frac{\varepsilon_{\mu\nu\lambda}}{8\pi} f_{abc} \partial_\mu n^a \partial_\nu n^b \partial_\lambda n^c , \tag{B3}$$

and, using Eq. (B2) and the contraction rule of the structure constants [44], we obtain

$$\partial_\mu J_\mu = \frac{\varepsilon_{\mu\nu\lambda}}{8\pi} f_{abc} f_{ade} \omega_\mu^d n^e \partial_\nu n^b \partial_\lambda n^c = \frac{\varepsilon_{\mu\nu\lambda}}{8\pi} \left[ \frac{2}{N}(\delta_{be}\delta_{cd} - \delta_{bd}\delta_{ce}) + d_{bea}d_{cda} - d_{bda}d_{cea} \right] \omega_\mu^d n^e \partial_\nu n^b \partial_\lambda n^c$$

$$= \frac{\varepsilon_{\mu\nu\lambda}}{8\pi} [d_{bea}d_{cda} - d_{bda}d_{cea}] \omega_\mu^d n^e \partial_\nu n^b \partial_\lambda n^c = \frac{\varepsilon_{\mu\nu\lambda}}{8\pi} \frac{N-2}{N} \left[ d_{cda}\partial_\nu n^a \partial_\lambda n^c - d_{bda}\partial_\nu n^b \partial_\lambda n^a \right] \omega_\mu^d$$

$$= 0 , \tag{B4}$$

Invoking gLL, the continuity equation $\partial_t J_t + \nabla \cdot \mathbf{J} = 0$ can be further written as a fractonic continuity equation:

$$\partial_t J_t = \frac{\varepsilon_{ij}}{4\pi} f_{abc} \partial_i \left[ n^a \partial_t n^b \partial_j n^c \right] = -\frac{\varepsilon_{ij}}{4\pi} f_{abc} \partial_i \left[ \partial_t n^a n^b \partial_j n^c \right] = -\frac{\varepsilon_{ij}}{2\pi} f_{abc} f_{ade} \partial_i \left[ \frac{\delta H}{\delta n^d} n^e n^b \partial_j n^c \right]$$

$$= -\frac{\varepsilon_{ij}}{2\pi} \left[ \frac{2}{N}(\delta_{be}\delta_{cd} - \delta_{bd}\delta_{ce}) + d_{bea}d_{cda} - d_{bda}d_{cea} \right] \partial_i \left[ \frac{\delta H}{\delta n^d} n^e n^b \partial_j n^c \right]$$

$$= -\frac{\varepsilon_{ij}}{2\pi} \left[ \frac{2}{N} \partial_i \left[ \frac{\delta H}{\delta n^c} n^b n^b \partial_j n^c \right] + (d_{bea}d_{cda} - d_{bda}d_{cea}) \partial_i \left[ \frac{\delta H}{\delta n^d} n^e n^b \partial_j n^c \right] \right]$$

$$= -\frac{\varepsilon_{ij}}{2\pi} \left[ \frac{2}{N} \frac{2(N-1)}{N} \partial_i \left[ \frac{\delta H}{\delta n^c} \partial_j n^c \right] + \frac{2(N-2)}{N} d_{cda} \partial_i \left[ \frac{\delta H}{\delta n^d} n^a \partial_j n^c \right] - \frac{(N-2)}{N} d_{bda} \partial_i \left[ \frac{\delta H}{\delta n^d} n^b \partial_j n^a \right] \right]$$

$$= -\frac{\varepsilon_{ij}}{2\pi} \left[ \frac{2}{N} \frac{2(N-1)}{N} \partial_i \left[ \frac{\delta H}{\delta n^c} \partial_j n^c \right] + \left( \frac{(N-2)}{N} \right)^2 \partial_i \left[ \frac{\delta H}{\delta n^d} \partial_j n^d \right] \right]$$

$$= -\frac{\varepsilon_{ij}}{2\pi} \partial_i \left[ \frac{\delta H}{\delta n^d} \partial_j n^d \right]$$

$$= -\frac{\varepsilon_{ij}}{2\pi} \partial_i \left[ \left[ \frac{\partial \mathcal{H}}{\partial n^d} - \partial_k \left( \frac{\partial \mathcal{H}}{\partial(\partial_k n^d)} \right) \right] \partial_j n^d \right] = -\frac{\varepsilon_{ij}}{2\pi} \partial_i \left[ \frac{\partial \mathcal{H}}{\partial n^d} \partial_j n^d + \frac{\partial \mathcal{H}}{\partial(\partial_k n^d)} \partial_j \partial_k n^d - \partial_k \left( \frac{\partial \mathcal{H}}{\partial(\partial_k n^d)} \partial_j n^d \right) \right]$$

$$= -\frac{\varepsilon_{ij}}{2\pi} \partial_i \left[ \partial_j \mathcal{H} - \partial_k \left( \frac{\partial \mathcal{H}}{\partial(\partial_k n^d)} \partial_j n^d \right) \right] = -\frac{\varepsilon_{ij}}{4\pi} \partial_i \left[ \partial_k \mathcal{H} \delta_{jk} - \partial_k \left( \frac{\partial \mathcal{H}}{\partial(\partial_k n^d)} \partial_j n^d \right) \right]$$

$$= -\frac{\varepsilon_{ij}}{2\pi} \partial_i \partial_k \left[ \mathcal{H} \delta_{jk} - \frac{\partial \mathcal{H}}{\partial(\partial_k n^d)} \partial_j n^d \right]$$

$$= -\frac{\varepsilon_{ij}}{2\pi} \partial_i \partial_k \sigma_{jk} . \tag{B5}$$

In the presence of dissipation, the field $n^a$ obeys the dynamics under the gLLG equation (3.6) and the continuity equation for the $CP^{N-1}$ fluid is modified as

$$\partial_t J_t = -\frac{\varepsilon_{ij}}{4\pi} f_{abc} \partial_i \left[ \partial_t n^a n^b \partial_j n^c \right] = -\frac{\varepsilon_{ij}}{2\pi} \partial_i \partial_k \sigma_{jk} + \alpha \frac{\varepsilon_{ij}}{4\pi} f_{abc} \partial_i \left[ \frac{\delta H}{\delta n^a} n^b \partial_j n^c \right] . \tag{B6}$$

## Appendix C: GMP algebra

In this section, we derive the Poisson bracket of the topological charge densities $\{J_t(\mathbf{r}), J_t(\mathbf{r}')\}$. From the Poisson bracket of the field $n^a$,

$$\{n^a(\mathbf{r}), n^b(\mathbf{r}')\} = 2 f_{abc} n^c(\mathbf{r}) \delta(\mathbf{r} - \mathbf{r}'),\tag{C1}$$

which corresponds to the commutation relation of the SU($N$) generators, the Poisson bracket of the field and its derivative is directly obtained as

$$\{\partial_i n^a(\mathbf{r}), n^b(\mathbf{r}')\} = 2 f_{abc} n^c(\mathbf{r}') \partial_i \delta(\mathbf{r} - \mathbf{r}').\tag{C2}$$

The Poisson bracket of the topological charge density and the field is computed by considering the following integral

$$\int_{\mathbf{r},\mathbf{r}'} \{J_t(\mathbf{r}), n^d(\mathbf{r}')\} g(\mathbf{r},\mathbf{r}') = \int_{\mathbf{r},\mathbf{r}'} \frac{\varepsilon_{ij}}{8\pi} f_{abc} \{n^a \partial_i n^b \partial_j n^c, n^d(\mathbf{r}')\} g = \int_{\mathbf{r},\mathbf{r}'} \frac{\varepsilon_{ij}}{8\pi} f_{abc} \left[ \{n^a, n^d(\mathbf{r}')\} \partial_i n^b \partial_j n^c + 2 n^a \partial_i n^b \{\partial_j n^c, n^d(\mathbf{r}')\} \right] g,$$

$$= \int_{\mathbf{r},\mathbf{r}'} \frac{\varepsilon_{ij}}{8\pi} f_{abc} \left[ 3 \{n^a, n^d(\mathbf{r}')\} \partial_i n^b \partial_j n^c g - 2 n^a \partial_i n^b \{n^c, n^d(\mathbf{r}')\} \partial_j g \right],\tag{C3}$$

where $g$ is an arbitrary function depending on $\mathbf{r}$ and $\mathbf{r}'$. Here, for the field variables, only the dependence on $\mathbf{r}'$ is explicitly written, and the argument $\mathbf{r}$ is omitted for brevity. One can show that the first term in the last expression of Eq. (C3) vanishes from the identities (A4). The second term becomes

$$\int_{\mathbf{r},\mathbf{r}'} \frac{\varepsilon_{ij}}{4\pi} f_{abc} n^c \partial_i n^b \{n^a, n^d(\mathbf{r}')\} \partial_j g = \int_{\mathbf{r},\mathbf{r}'} \frac{\varepsilon_{ij}}{2\pi} f_{abc} f_{ade} n^c \partial_i n^b n^e \delta(\mathbf{r} - \mathbf{r}') \partial_j g$$

$$= \int_{\mathbf{r},\mathbf{r}'} \frac{\varepsilon_{ij}}{2\pi} \left[ \frac{2}{N} (\delta_{be}\delta_{cd} - \delta_{bd}\delta_{ce}) + d_{bea}d_{cda} - d_{bda}d_{cea} \right] n^c \partial_i n^b n^e \delta(\mathbf{r} - \mathbf{r}') \partial_j g$$

$$= \int_{\mathbf{r},\mathbf{r}'} \frac{\varepsilon_{ij}}{2\pi} \left[ -\frac{2}{N} \frac{2(N-1)}{N} \partial_i n^d + \frac{N-2}{N} d_{cda} \partial_i n^a n^c - \frac{2(N-2)}{N} d_{bda} \partial_i n^b n^a \right] \delta(\mathbf{r} - \mathbf{r}') \partial_j g$$

$$= -\int_{\mathbf{r},\mathbf{r}'} \frac{\varepsilon_{ij}}{2\pi} \partial_i n^d \delta(\mathbf{r} - \mathbf{r}') \partial_j g$$

$$= \int_{\mathbf{r},\mathbf{r}'} \frac{\varepsilon_{ij}}{2\pi} \partial_i n^d \partial_j \delta(\mathbf{r} - \mathbf{r}') g.\tag{C4}$$

Thus, we have

$$\{J_t(\mathbf{r}), n^d(\mathbf{r}')\} = \frac{\varepsilon_{ij}}{2\pi} \partial_i n^d(\mathbf{r}) \partial_j \delta(\mathbf{r} - \mathbf{r}').\tag{C5}$$

For the Poisson bracket of the topological charge densities, we similarly compute the following quantity

$$\int_{\mathbf{r},\mathbf{r}'} \{J_t(\mathbf{r}), J_t(\mathbf{r}')\} g(\mathbf{r},\mathbf{r}') = \int_{\mathbf{r},\mathbf{r}'} \left[ \{n^d, J_t(\mathbf{r}')\} \frac{\partial J_t}{\partial n^d} + \{\partial_k n^d, J_t(\mathbf{r}')\} \frac{\partial J_t}{\partial \partial_k n^d} \right] g$$

$$= \int_{\mathbf{r},\mathbf{r}'} \{n^d, J_t(\mathbf{r}')\} \left[ \frac{\partial J_t}{\partial n^d} - \partial_k \frac{\partial J_t}{\partial \partial_k n^d} \right] g - \int_{\mathbf{r},\mathbf{r}'} \{n^d, J_t(\mathbf{r}')\} \frac{\partial J_t}{\partial \partial_k n^d} \partial_k g\tag{C6}$$

The first term in the last line of Eq. (C6) gives

$$\int_{\mathbf{r},\mathbf{r}'} \{n^d, J_t(\mathbf{r}')\} \left[ \frac{\partial J_t}{\partial n^d} - \partial_k \frac{\partial J_t}{\partial \partial_k n^d} \right] g = -\int_{\mathbf{r},\mathbf{r}'} \frac{\varepsilon_{ij}}{2\pi} \partial_i' n^d(\mathbf{r}') \partial_j' \delta(\mathbf{r} - \mathbf{r}') \left[ \frac{\partial J_t}{\partial n^d} - \partial_k \frac{\partial J_t}{\partial \partial_k n^d} \right] g$$

$$= \int_{\mathbf{r},\mathbf{r}'} \frac{\varepsilon_{ij}}{2\pi} \partial_i' n^d(\mathbf{r}') \delta(\mathbf{r} - \mathbf{r}') \left[ \frac{\partial J_t}{\partial n^d} - \partial_k \frac{\partial J_t}{\partial \partial_k n^d} \right] \partial_j' g$$

$$= \int_{\mathbf{r},\mathbf{r}'} \frac{\varepsilon_{ij}}{2\pi} \partial_i n^d(\mathbf{r}) \delta(\mathbf{r} - \mathbf{r}') \left[ \frac{\partial J_t}{\partial n^d} - \partial_k \frac{\partial J_t}{\partial \partial_k n^d} \right] \partial_j' g$$

$$= \int_{\mathbf{r},\mathbf{r}'} \frac{\varepsilon_{ij}}{2\pi} \left[ \partial_i n^d \frac{\partial J_t}{\partial n^d} + \partial_i \partial_k n^d \frac{\partial J_t}{\partial \partial_k n^d} \right] \delta(\mathbf{r} - \mathbf{r}') \partial_j' g + \int_{\mathbf{r},\mathbf{r}'} \frac{\varepsilon_{ij}}{2\pi} \partial_i n^d \frac{\partial J_t}{\partial \partial_k n^d} \partial_k \left[ \delta(\mathbf{r} - \mathbf{r}') \partial_j' g \right]$$

$$= \int_{\mathbf{r},\mathbf{r}'} \frac{\varepsilon_{ij}}{2\pi} \partial_i J_t \delta(\mathbf{r} - \mathbf{r}') \partial_j' g + \int_{\mathbf{r},\mathbf{r}'} \frac{\varepsilon_{ij}}{2\pi} \partial_i n^d \frac{\partial J_t}{\partial \partial_k n^d} \partial_k \left[ \delta(\mathbf{r} - \mathbf{r}') \partial_j' g \right].\tag{C7}$$

Meanwhile, the second term of Eq. (C6) is

$$-\int_{\mathbf{r},\mathbf{r}'}\{n^d,J_t(\mathbf{r}')\}\frac{\partial J_t}{\partial\partial_k n^d}\partial_k g = \int_{\mathbf{r},\mathbf{r}'}\frac{\varepsilon_{ij}}{2\pi}\partial_i n^d(\mathbf{r}')\partial_j'\delta(\mathbf{r}-\mathbf{r}')\frac{\partial J_t}{\partial\partial_k n^d}\partial_k g = -\int_{\mathbf{r},\mathbf{r}'}\frac{\varepsilon_{ij}}{2\pi}\partial_i n^d\delta(\mathbf{r}-\mathbf{r}')\frac{\partial J_t}{\partial\partial_k n^d}\partial_j'\partial_k g. \tag{C8}$$

Taking the sum of Eqs. (C7) and (C8), we can express Eq. (C6) as

$$\int_{\mathbf{r},\mathbf{r}'}\{J_t(\mathbf{r}),J_t(\mathbf{r}')\}g(\mathbf{r},\mathbf{r}') = \int_{\mathbf{r},\mathbf{r}'}\frac{\varepsilon_{ij}}{2\pi}\partial_i J_t\delta(\mathbf{r}-\mathbf{r}')\partial_j'g + \int_{\mathbf{r},\mathbf{r}'}\frac{\varepsilon_{ij}}{2\pi}\partial_i n^d\frac{\partial J_t}{\partial\partial_k n^d}\partial_k\delta(\mathbf{r}-\mathbf{r}')\partial_j'g$$

$$= \int_{\mathbf{r},\mathbf{r}'}\frac{\varepsilon_{ij}}{2\pi}\partial_i J_t\partial_j\delta(\mathbf{r}-\mathbf{r}')g + \int_{\mathbf{r},\mathbf{r}'}\frac{\varepsilon_{ij}}{2\pi}\partial_i n^d\frac{\partial J_t}{\partial\partial_k n^d}\partial_j\partial_k\delta(\mathbf{r}-\mathbf{r}')g$$

$$= \int_{\mathbf{r},\mathbf{r}'}\frac{\varepsilon_{ij}}{2\pi}\partial_i J_t\partial_j\delta(\mathbf{r}-\mathbf{r}')g + \int_{\mathbf{r},\mathbf{r}'}\frac{\varepsilon_{ij}}{2\pi}\frac{\varepsilon_{kl}}{4\pi}f_{adc}n^a\partial_l n^c\partial_i n^d\partial_j\partial_k\delta(\mathbf{r}-\mathbf{r}')g$$

$$= \int_{\mathbf{r},\mathbf{r}'}\frac{\varepsilon_{ij}}{2\pi}\partial_i J_t\partial_j\delta(\mathbf{r}-\mathbf{r}')g + \int_{\mathbf{r},\mathbf{r}'}\frac{f_{adc}}{8\pi^2}\left[n^a\partial_x n^c\partial_y n^d\partial_x\partial_y\delta(\mathbf{r}-\mathbf{r}')+n^a\partial_y n^c\partial_x n^d\partial_y\partial_x\delta(\mathbf{r}-\mathbf{r}')\right]g$$

$$= \int_{\mathbf{r},\mathbf{r}'}\frac{\varepsilon_{ij}}{2\pi}\partial_i J_t\partial_j\delta(\mathbf{r}-\mathbf{r}')g. \tag{C9}$$

Therefore, the Poisson bracket of the topological charge densities is given as the long-wavelength limit of the GMP algebra.

$$\{J_t(\mathbf{r}),J_t(\mathbf{r}')\} = \frac{\varepsilon_{ij}}{2\pi}\partial_i J_t(\mathbf{r})\partial_j\delta(\mathbf{r}-\mathbf{r}'). \tag{C10}$$

### Appendix D: CP² skyrmion

The spin-1 state is spanned by three angular momentum states $|1\rangle,|0\rangle,|-1\rangle$, or by the Cartesian basis $\{|x\rangle,|y\rangle,|z\rangle\}$ [6, 10]:

$$|x\rangle = i(|1\rangle-|-1\rangle)/\sqrt{2}, \quad |y\rangle = (|1\rangle+|-1\rangle)/\sqrt{2}, \quad |z\rangle = -i|0\rangle. \tag{D1}$$

The eight generators of SU(3), $\lambda_a$ ($a=1,\cdots,8$) are related to the three magnetic and five quadrupole moments of spin-1 through

$$\begin{pmatrix}\lambda^7\\\lambda^5\\\lambda^2\end{pmatrix} = \begin{pmatrix}S^x\\-S^y\\S^z\end{pmatrix}, \quad -\begin{pmatrix}\lambda^3\\\lambda^8\\\lambda^1\\\lambda^4\\\lambda^6\end{pmatrix} = \begin{pmatrix}Q^{x^2-y^2}\\Q^{3z^2-r^2}\\Q^{xy}\\Q^{xz}\\Q^{yz}\end{pmatrix} = \begin{pmatrix}(S^x)^2-(S^y)^2\\\frac{1}{\sqrt{3}}[\mathbf{S}\cdot\mathbf{S}-3(S^z)^2]\\S^xS^y+S^yS^x\\S^zS^x+S^xS^z\\S^yS^z+S^zS^y\end{pmatrix}, \tag{D2}$$

For a given $\mathbf{z}$, the magnetic dipole and quadrupole components can be worked out as $n^a = \mathbf{z}^\dagger\lambda_a\mathbf{z} = z_i^*(\lambda_a)_{ij}z_j$, with the components $z_j$ given in the Cartesian basis $|\mathbf{z}\rangle = \sum_{j=x,y,z}z_j|j\rangle$.

For $\mathbf{z}$ given in the exact solution of the CP² skyrmion [Eq. (4.5)], the magnetic dipole and quadrupole moments are

$$\langle\mathbf{S}\rangle = \frac{i}{\Delta}\left(\xi^{2|n|}(\mathbf{u}\times\mathbf{u}^*)+\xi^{|n|}r^{|n|}e^{in\theta}(\mathbf{v}\times\mathbf{u}^*)+\xi^{|n|}r^{|n|}e^{-in\theta}(\mathbf{u}\times\mathbf{v}^*)+r^{2|n|}(\mathbf{v}\times\mathbf{v}^*)\right),$$

$$\langle Q^{x^2-y^2}\rangle = -\frac{1}{\Delta}\left(\xi^{2|n|}(u_1^*u_1-u_2^*u_2)+\xi^{|n|}r^{|n|}e^{in\theta}(u_1^*v_1-u_2^*v_2)+\xi^{|n|}r^{|n|}e^{-in\theta}(v_1^*u_1-v_2^*u_2)+r^{2|n|}(v_1^*v_1-v_2^*v_2)\right),$$

$$\langle Q^{3z^2-r^2}\rangle = -\frac{1}{\Delta\sqrt{3}}\left(\xi^{2|n|}(|\mathbf{u}|^2-3u_3^*u_3)+\xi^{|n|}r^{|n|}e^{in\theta}(\mathbf{u}^*\cdot\mathbf{v}-3u_3^*v_3)+\xi^{|n|}r^{|n|}e^{-in\theta}(\mathbf{v}^*\cdot\mathbf{u}-3v_3^*u_3)+r^{2|n|}(|\mathbf{v}|^2-3v_3^*v_3)\right),$$

$$\langle Q^{xy}\rangle = -\frac{1}{\Delta}\left(\xi^{2|n|}(u_2^*u_1+u_1^*u_2)+\xi^{|n|}r^{|n|}e^{in\theta}(u_2^*v_1+u_1^*v_2)+\xi^{|n|}r^{|n|}e^{-in\theta}(v_2^*u_1+v_1^*u_2)+r^{2|n|}(v_2^*v_1+v_1^*v_2)\right),$$

$$\langle Q^{xz}\rangle = -\frac{1}{\Delta}\left(\xi^{2|n|}(u_3^*u_1+u_1^*u_3)+\xi^{|n|}r^{|n|}e^{in\theta}(u_3^*v_1+u_1^*v_3)+\xi^{|n|}r^{|n|}e^{-in\theta}(v_3^*u_1+v_1^*u_3)+r^{2|n|}(v_3^*v_1+v_1^*v_3)\right),$$

$$\langle Q^{yz}\rangle = -\frac{1}{\Delta}\left(\xi^{2|n|}(u_3^*u_2+u_2^*u_3)+\xi^{|n|}r^{|n|}e^{in\theta}(u_3^*v_2+u_2^*v_3)+\xi^{|n|}r^{|n|}e^{-in\theta}(v_3^*u_2+v_2^*u_3)+r^{2|n|}(v_3^*v_2+v_2^*v_3)\right), \tag{D3}$$

where $\Delta = \xi^{2|n|}+r^{2|n|}$. The scalar spin chirality is given by

$$\langle\mathbf{S}\rangle\cdot\partial_x\langle\mathbf{S}\rangle\times\partial_y\langle\mathbf{S}\rangle = \frac{2n|n|(r\xi)^{2|n|}}{r^2\Delta^3}\left[r^{2|n|}|\mathbf{u}\cdot(\mathbf{v}\times\mathbf{v}^*)|^2+\xi^{2|n|}|\mathbf{v}\cdot(\mathbf{u}\times\mathbf{u}^*)|^2\right.$$

$$\left.-(r\xi)^{|n|}e^{-in\theta}(\mathbf{v}\times\mathbf{v}^*)\cdot((\mathbf{u}\times\mathbf{v}^*)\times(\mathbf{u}\times\mathbf{u}^*))+(r\xi)^{|n|}e^{in\theta}(\mathbf{v}\times\mathbf{v}^*)\cdot((\mathbf{v}\times\mathbf{u}^*)\times(\mathbf{u}\times\mathbf{u}^*))\right]. \tag{D4}$$

The norm squared of the magnetic dipole moment is worked out to be

$$\langle \mathbf{S} \rangle \cdot \langle \mathbf{S} \rangle = -\frac{1}{\Delta^2}\Big[ \xi^{4|n|}(\mathbf{u}\times\mathbf{u}^*)\cdot(\mathbf{u}\times\mathbf{u}^*) + r^{4|n|}(\mathbf{v}\times\mathbf{v}^*)\cdot(\mathbf{v}\times\mathbf{v}^*) + 2\xi^{2|n|}r^{2|n|}\left(|\mathbf{u}\cdot\mathbf{v}|^2 - 1\right)$$

$$+2\xi^{3|n|}r^{|n|}\left(e^{in\theta}(\mathbf{u}\cdot\mathbf{v})(\mathbf{u}^*\cdot\mathbf{u}^*) + e^{-in\theta}(\mathbf{u}\cdot\mathbf{u})(\mathbf{u}^*\cdot\mathbf{v}^*)\right) + 2\xi^{|n|}r^{3|n|}\left(e^{in\theta}(\mathbf{v}\cdot\mathbf{v})(\mathbf{u}^*\cdot\mathbf{v}^*) + e^{-in\theta}(\mathbf{u}\cdot\mathbf{v})(\mathbf{v}^*\cdot\mathbf{v}^*)\right)$$

$$+2\xi^{2|n|}r^{2|n|}\left(e^{i2n\theta}(\mathbf{v}\cdot\mathbf{v})(\mathbf{u}^*\cdot\mathbf{u}^*) + e^{-i2n\theta}(\mathbf{u}\cdot\mathbf{u})(\mathbf{v}^*\cdot\mathbf{v}^*)\right)\Big]. \tag{D5}$$

## Appendix E: Energy minimization for the CP$^2$ skyrmion

By decomposing the complex vectors into their real and imaginary parts, $\mathbf{u} = \mathbf{u}_r + i\mathbf{u}_i$ and $\mathbf{v} = \mathbf{v}_r + i\mathbf{v}_i$, the quadratic term in the energy (4.8) can be expressed as

$$\kappa \int_{\mathbf{r}} \langle \mathbf{S} \rangle \cdot \langle \mathbf{S} \rangle = \kappa \int \frac{2\pi}{(\xi^{2|n|}+r^{2|n|})^2}\left[\xi^{4|n|}4|\mathbf{u}_r\times\mathbf{u}_i|^2 + r^{4|n|}4|\mathbf{v}_r\times\mathbf{v}_i|^2 - 2\xi^{2|n|}r^{2|n|}\left(|\mathbf{u}\cdot\mathbf{v}|^2 - 1\right)\right] r\,dr. \tag{E1}$$

For $\kappa > 0$, one can show that the vectors minimizing the energy functional are given by

$$\mathbf{u} = e^{i\frac{\alpha}{2}}\hat{\mathbf{e}}, \quad \mathbf{v} = e^{i\frac{\beta}{2}}\hat{\mathbf{e}}, \tag{E2}$$

with two arbitrary phases $\alpha, \beta$, and an arbitrary real unit vector $\hat{\mathbf{e}}$. In the case of $\kappa < 0$, there exists no pair of constant orthonormal vectors $(\mathbf{u}, \mathbf{v})$ minimizing all terms in the integrand of Eq. (E1). Since the characteristic length $\xi$ is much smaller than the system size, the contribution of the term proportional to $\xi^{4|n|}$ to the total energy can be neglected. From the second and third terms of Eq. (E1), we find the following relations

$$|\mathbf{v}_r| = |\mathbf{v}_i| = \frac{1}{\sqrt{2}}, \quad \mathbf{v}_r\cdot\mathbf{v}_i = 0, \quad \mathbf{u} = 2e^{i\varphi}\mathbf{v}_r\times\mathbf{v}_i \tag{E3}$$

where $\varphi$ is an arbitrary phase. The quartet of real vectors $\{\mathbf{v}_r, \mathbf{v}_i, \mathbf{u}_r, \mathbf{u}_i\}$ forms a rigid body which is parameterized by SO(3) rotation, and the compleax vector $\mathbf{u}$ has U(1) degeneracy for given $\mathbf{v}_r$ and $\mathbf{v}_i$. Thus the configuration is parameterized by the SO(3) rotation $\mathscr{R}(\alpha,\beta,\gamma) = e^{\alpha L_z}e^{\beta L_y}e^{\gamma L_z}$, where $L_i$ are the generators of SO(3), and the U(1) rotation $e^{i\varphi}$ as

$$\mathbf{v}_r = \mathscr{R}(\alpha,\beta,\gamma)\begin{pmatrix}\frac{1}{\sqrt{2}}\\0\\0\end{pmatrix} = \frac{1}{\sqrt{2}}\begin{pmatrix}\cos\alpha\cos\beta\cos\gamma - \sin\alpha\sin\gamma\\\cos\beta\cos\gamma\sin\alpha + \cos\alpha\sin\gamma\\-\cos\gamma\sin\beta\end{pmatrix},$$

$$\mathbf{v}_i = \mathscr{R}(\alpha,\beta,\gamma)\begin{pmatrix}0\\\frac{1}{\sqrt{2}}\\0\end{pmatrix} = \frac{1}{\sqrt{2}}\begin{pmatrix}-\cos\alpha\cos\beta\sin\gamma - \cos\gamma\sin\alpha\\-\cos\beta\sin\alpha\sin\gamma + \cos\alpha\cos\gamma\\\sin\beta\sin\gamma\end{pmatrix},$$

$$\mathbf{v} = \frac{e^{-i\gamma}}{\sqrt{2}}\begin{pmatrix}\cos\alpha\cos\beta - i\sin\alpha\\\sin\alpha\cos\beta + i\cos\alpha\\-\sin\beta\end{pmatrix}, \quad \mathbf{u} = e^{i\varphi}\begin{pmatrix}\cos\alpha\sin\beta\\\sin\alpha\sin\beta\\\cos\beta\end{pmatrix}. \tag{E4}$$

Employing this solution, the magnetic dipole moments from Eq. (D3) are expressed as

$$\langle S^x \rangle = \frac{-\sqrt{2}\xi^n r^n(\sin\alpha\sin\psi + \cos\alpha\cos\beta\cos\psi) + r^{2n}\cos\alpha\sin\beta}{\xi^{2|n|}+r^{2|n|}},$$

$$\langle S^y \rangle = \frac{\sqrt{2}\xi^n r^n(\cos\alpha\sin\psi - \sin\alpha\cos\beta\cos\psi) + r^{2n}\sin\alpha\sin\beta}{\xi^{2|n|}+r^{2|n|}},$$

$$\langle S^z \rangle = \frac{\sqrt{2}\xi^n r^n\sin\beta\cos\psi + r^{2n}\cos\beta}{\xi^{2|n|}+r^{2|n|}}, \tag{E5}$$

similarly, the magnetic quadrupole moments can be recast as follows:

$$\langle Q^{x^2-y^2}\rangle = \frac{\sin\beta}{2(r^{2|n|}+\xi^{2|n|})}\left[(r^{2|n|}-2\xi^{2|n|})\cos 2\alpha\sin\beta - 2\sqrt{2}(r\xi)^{|n|}(\cos 2\alpha\cos\beta\cos\psi + \sin 2\alpha\sin\phi)\right],$$

$$\langle Q^{3z^2-r^2}\rangle = \frac{-1}{4\sqrt{3}(r^{2|n|}+\xi^{2|n|})}\left[(r^{2|n|}-2\xi^{2|n|})(1+3\cos 2\beta) + 6\sqrt{2}(r\xi)^{|n|}\cos\psi\sin 2\beta\right],$$

$$\langle Q^{xy}\rangle = \frac{\sin\beta}{2(r^{2|n|}+\xi^{2|n|})}\left[(r^{2|n|}-2\xi^{2|n|})\sin 2\alpha\sin\beta - 2\sqrt{2}(r\xi)^{|n|}(\cos\beta\cos\psi\sin 2\alpha - \cos 2\alpha\sin\psi)\right],$$

$$\langle Q^{xz}\rangle = \frac{1}{2(r^{2|n|}+\xi^{2|n|})}\left[(r^{2|n|}-2\xi^{2|n|})\cos\alpha\sin 2\beta - 2\sqrt{2}(r\xi)^{|n|}(\cos\alpha\cos 2\beta\cos\psi + \cos\beta\sin\alpha\sin\psi)\right],$$

$$\langle Q^{yz}\rangle = \frac{1}{2(r^{2|n|}+\xi^{2|n|})}\left[(r^{2|n|}-2\xi^{2|n|})\sin\alpha\sin 2\beta - 2\sqrt{2}(r\xi)^{|n|}(\sin\alpha\cos 2\beta\cos\psi - \cos\beta\cos\alpha\sin\psi)\right], \quad \text{(E6)}$$

where $\psi = n\theta - \varphi - \gamma$.

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
