# Peer review of "Dynamics of $\mathrm{CP}^{N-1}$ skyrmions"

_SciPost Physics_

## Round 1 · Referee Report · Anonymous (Referee 1) · 2025-11-28

Strengths

The paper treats $CP^{N-1}$ generically with a general Hamiltonian, which is quite elegant, and as explained in the introduction, it covers several physically interesting cases: $N=2,3,4$.

Weaknesses

Only the BPS equation is used to find solutions and it is used incompatibly with a potential in the case of the "ferromagnetic" solutions, see the report.

Report

This paper is concerned with the dynamics of $CP^{N-1}$ skyrmions by extending the known Landau-Lifshitz (LL) or Landau-Lifshitz-Gilbert (LLG) equation to their generalizations for the $CP^{N-1}$ skyrmions (called gLL and gLLG equations in the paper).
The continuity equation, here derived as the divergence of the topological current, is satisfied for vanishing Gilbert damping.
It is also rewritten as a fractonic conservation law.
The authors find a modified continuity equation which holds also with nonvanishing Gilbert damping, but the dipole moments are not conserved.
A general solution of the $CP^2$ case is found using the BPS equation (4.4), which has two moduli, which are two complex 3-vectors ($u$ and $v$) with an orthogonality constraint as well as a size modulus, so 12 real moduli in total.
In order to fix the moduli, a potential (4.8) is introduced, which is a sum of squares of three of the fields: $n^5$, $-n^7$ and $n^2$, so the potential is
\[\kappa[(n^5)^2 + (n^7)^2 + (n^2)^2].\]
Two different minimizations of the potential are found according to
the sign of $\kappa$.
The author coin the positive $\kappa>0$ case "quadrupolar $CP^2$" and the negative $\kappa<0$ case "ferromagnetic $CP^2$".
Some speculations of the $CP^2$ skyrmions exhibiting the topological Hall effect are made.

There are some issues with this paper that the authors need to address, see the requested changes below.

Requested changes

1) The $CP^{N-1}$ and in particular $CP^2$ skyrmion solutions are found using the BPS equation (4.4), which is derived in [15] for the case without a potential. In principle, such solution cannot be used after a potential is introduced, although in some limiting cases, one could study infinitesimally small potentials, but one would need to be careful.

In the first case ($\kappa>0$), the potential is minimized and vanishes after minimization, so the Bogomol'nyi completion in (4.1)-(4.2) is still valid: this "quadrupolar" solution is merely a subset of the full BPS solution (a subset of the moduli space of solutions).

In the second case ($\kappa<0$), the potential does not vanish. Therefore, using first the BPS equation and then energy minimization on only the potential term, does not guarantee that the minimum of the total energy function is obtained. There may well be a non-BPS solution that has even lower energy, than the BPS solution the authors have found. The non-BPS solution may also lift some or even all of the moduli (called $\alpha$, $\beta$, $\gamma$ and $\varphi$ on page 4). Or there may exist a different Bogomol'nyi completion, which will yield the correct BPS equation for the case at hand.

A correct solution should be found or some convincing arguments should explain why the solution obtained is indeed a minimizer of the total energy functional.

2) The specific Hamiltonian is not introduced clearly in the paper, but from the (4.2) is seems that only the kinetic term (Dirichlet energy) is used. This is a problem, since the solitons are not stabilized in size, like magnetic skyrmions are (that are stabilized by a DMI term counteracting a Zeeman energy). Indeed, the solution in (4.5) contains a size modulus: these solutions are hence sigma-model lumps and not skyrmions (although they have the exact same topology).

It must be clarified, why these scale-invariant solutions are of any interest to condensed matter systems; probably the answer is that they are not. It would be better to introduce a generalized DMI term, like that of [11] as well as a potential. Such terms may change some of the physics or conclusions drawn in this paper.

3) The authors have probably not made mistakes, but it is difficult to check the signs in the equations, since the authors are too sloppy with raised and lowered Minkowski indices: $\mu$, $\nu$, ... where $\mu=0,1,2$. The affected equations are: (2.5), (2.7), $\partial_\mu J_\mu=0$ above (3.1) which leads to (3.1) and is crucial for the work, (B1), (B3), (B4). If no dynamics and only Euclidean space was used, the double lowered spacetime indices would be acceptable, but the paper is about dynamics. Indeed the current conservation in Minkowski space reads

$$\partial_\mu J^\mu = -\partial_0 J_0 + \partial_i J_i = 0.$$

4) The formal derivation of the Landau-Lifshitz equation in appendix A needs more detail. What are the assumptions of the auxiliary $u$-dependence of the fields? (no dependence would make (A1) vanish). Performing the variation on (A1) I get:

$$\frac32\int_0^1 du f_{abc}(\partial_u n^a) (\partial_t n^b) \delta n^c + {\rm boundary\ term}, $$
which, however, does not straightforwardly integrate to the expression
$$\frac12(\partial_t n^a) n^b \delta n^c.$$

Recommendation

Ask for major revision

  • validity: poor
  • significance: ok
  • originality: good
  • clarity: low
  • formatting: good
  • grammar: good

Author:  Seungho Lee  on 2025-12-07  [id 6113]

(in reply to Report 1 on 2025-11-28)
Category:
answer to question

We thank the referee for a careful and insightful reading of our manuscript and for the constructive comments that help to clarify several important aspects of our work. We address the four requested changes in detail below.

Referee 1: 1) The $\mathrm{CP}^{N-1}$ and in particular $\mathrm{CP}^2$ skyrmion solutions are found using the BPS equation (4.4), which is derived in [15] for the case without a potential. In principle, such solution cannot be used after a potential is introduced, although in some limiting cases, one could study infinitesimally small potentials, but one would need to be careful.

In the first case ($\kappa \gt 0$), the potential is minimized and vanishes after minimization, so the Bogomol'nyi completion in (4.1)–(4.2) is still valid: this "quadrupolar" solution is merely a subset of the full BPS solution (a subset of the moduli space of solutions).

In the second case ($\kappa \lt 0$), the potential does not vanish. Therefore, using first the BPS equation and then energy minimization on only the potential term, does not guarantee that the minimum of the total energy function is obtained. There may well be a non–BPS solution that has even lower energy than the BPS solution the authors have found. The non–BPS solution may also lift some or even all of the moduli (called $\alpha, \beta, \gamma$ and $\varphi$ on page 4). Or there may exist a different Bogomol'nyi completion, which will yield the correct BPS equation for the case at hand.

A correct solution should be found or some convincing arguments should explain why the solution obtained is indeed a minimizer of the total energy functional.

Response: The referee correctly points out that the BPS equation (4.4) is derived for the pure sigma model, and that inclusion of the potential term with nonzero $\kappa$ can modify the form of the true energy-minimizing configuration. In accordance with the rigorous view advocated by the referee, we have clarified our discussions in various places.

For $\kappa \gt 0$, the potential term (4.8) is minimized by configurations with strictly vanishing local magnetization, and the BPS-type solution again gives the true energy-minimizing solution provided the magnetic moments are zero everywhere. This gives rise to the formation of quadrupolar skyrmion solution we have presented in the text. We have clearly stated these points in the revised manuscript so that readers should have no confusion in regard to the validity of the BPS-type solutions in the case of $\kappa \gt 0$ anisotropy.

For $\kappa \lt 0$, as the referee has rightly pointed out, the potential term is not fully minimized by the BPS-type solution and the energy analysis we have performed remains approximate. We argue, in accordance with the referee comment, that this should remain a good approximation for sufficiently small $\kappa \lt 0$ and does not violate the spirit of the variational calculation we have pursued in this article. The full classification of non-BPS solutions and investigating the subspace of the moduli space are left for future work.

In response to the referee’s suggestion, we have added a paragraph to Sec. IV of the manuscript:
"For $\kappa \lt 0$, the anistropy energy can be written, up to a constant, as $|\kappa| \sum_a (Q^a )^2$ where $a$ runs over the five quadrupolar components. To minimize the anistropy energy fully, one would like to seek solutions for which all $Q^a =0$. This, however, implies that one is left with only the magnetic texture $\bf S$, which can only give rise to the CP$^1$ skymion charge. A more natural solution is the one where the quadrupole moments are suppressed in most parts of space but not in all, in accordance with the picture of our ferromagnetic CP$^2$ skyrmion solution. For sufficiently small $\kappa \lt 0$, the ferromagnetic solution found within the manifold of BPS-type skyrmions should remain a good approximation to the true energy-minimizing solution."

Referee 1: 2) The specific Hamiltonian is not introduced clearly in the paper, but from the (4.2) is seems that only the kinetic term (Dirichlet energy) is used. This is a problem, since the solitons are not stabilized in size, like magnetic skyrmions are (that are stabilized by a DMI term counteracting a Zeeman energy). Indeed, the solution in (4.5) contains a size modulus: these solutions are hence sigma–model lumps and not skyrmions (although they have the exact same topology).

It must be clarified, why these scale–invariant solutions are of any interest to condensed matter systems; probably the answer is that they are not. It would be better to introduce a generalized DMI term, like that of [11] as well as a potential. Such terms may change some of the physics or conclusions drawn in this paper.

Response: The referee is right in saying that in Sec. IV we focus on the scale-invariant CP$^2$ sigma model with an anisotropy term, and that we do not include a Dzyaloshinskii–Moriya interaction (DMI) or Zeeman field that would energetically fix the radius of the CP$^2$ skyrmion as would be the case in the more realistic calculations of chiral magnets. The solutions (4.5) indeed describe skyrmions with the size modulus remaining as a free parameter. We must emphasize, however, that the main interest of our work is understanding whether the topological Hall effect can arise even as a matter of principle in CP$^2$ magnets where the skyrmions are Hund-coupled to itinerant electrons. As such, fixing the radius of the skyrmion with more realistic calculations will not be of imminent concern.

In accordance with the referee comment, we clarify that:

• Our main results— fractonic continuity equation, its modification by Gilbert damping, and the GMP algebra for the CP$^{N-1}$ topological density—hold for \emph{arbitrary} local Hamiltonians $H[n, \partial n]$, in particular in regard to whether the DMI and/or Zeeman terms are included.

• Sec. IV uses the scale-invariant CP$^2$ sigma model and its solutions only as a tractable platform to analyze how a CP$^2$ skyrmion can carry a nonzero embedded CP$^1$ charge, which is the quantity of relevance in understanding the topological Hall effect in CP$^2$ fluid coupled to mobile electrons. According to our variational analysis, we conclude there can be two different types of CP$^2$ skyrmions - one quadrupolar and one ferromagnetic - and only the latter can result in topological Hall effect. This conclusion remains insensitive to the precise value of the skyrmion radius since we are not at the moment interested in estimating the actual Hall angle arising from the electron trajectory passing through the CP$^2$ skyrmion. We added comments in regard to the heuristic and qualitative nature of the variational calculations and the conclusions reached in regard to the topological Hall effect, and indicate that a fully realistic description of specific materials including generalized DMI and Zeeman terms remains as a future study.

In response to the referee’s request, we have added the following paragraph to Summary as follows:
"In the future, a more thorough analysis of the stability of the CP$^{N-1}$ skyrmion solution in the presence of both the generalized Dzyaloshinskii–Moriya interaction [10, 11], frustrating exchange interaction [12], as well as the Zeeman term may be desired. On the other hand, our central findings -- fractonic continuity equation and its modification by the Gilbert damping, as well as the GMP algebra obeyed by the CP$^{N-1}$ topological density -- remain valid regardless of the specific terms entering in the Hamiltonian. Our discussion of the topological Hall effect for the ferromagnetic CP$^2$ skyrmion remains independent of the precise determination of the skyrmion radius $\xi$, which follows from the competition among various magnetic interactions and Zeeman energy."

Referee 1: 3) The authors have probably not made mistakes, but it is difficult to check the signs in the equations, since the authors are too sloppy with raised and lowered Minkowski indices: $\mu, \nu, \ldots$ where $\mu = 0,1,2$. The affected equations are: (2.5), (2.7), $\partial_\mu J^\mu = 0$ above (3.1) which leads to (3.1) and is crucial for the work, (B1), (B3), (B4). If no dynamics and only Euclidean space was used, the double lowered spacetime indices would be acceptable, but the paper is about dynamics. Indeed the current conservation in Minkowski space reads
\begin{equation}
\partial_\mu J^\mu = -\partial_0 J_0 + \partial_i J_i = 0. \nonumber
\end{equation}

Response: We thank the referee for pointing out the confusing use of raised and lowered spacetime indices in Sec.\ II–III and Appendix B. In the revised manuscript we have adopted a consistent Minkowski convention $x^\mu = (t, x, y)$, metric $\eta_{\mu\nu} = \mathrm{diag}(1,-1,-1)$. Following the referee's opinion, we have modified Eq.~(2.5) as follows:"
\begin{align}
J^\mu = \frac{1}{8\pi} \epsilon^{\mu\nu\lambda}f_{abc} n^a \partial_\nu n^b \partial_\lambda n^c = \frac{1}{2\pi} \epsilon^{\mu\nu\lambda} \partial_\nu a_\lambda
\end{align}
where $\mu = t,x,y$, $a_\mu = - i {\bf z}^\dagger \partial_\mu {\bf z}$, and we adopt the metric signature $(+,-,-)$."
Eqs.~(2.7), (B1), (B3), and (B4) have been revised correspondingly. We have also modified the continuity equation to the correct form
\begin{align}
\partial_\mu J^\mu = 0\,.
\end{align}
The main result [Eq.~(3.1)] does not require modification, since under the metric signature $(+,-,-)$,
\begin{align}
\partial_t J^t + \partial_i \partial_j J^{ij} = \partial_t J_t + \partial_i \partial_j J_{ij} = 0\,.
\end{align}

We believe this notational cleanup significantly improves readability and removes the ambiguity in the sign structure emphasized by the referee.

Referee 1: 4) The formal derivation of the Landau–Lifshitz equation in appendix A needs more detail. What are the assumptions of the auxiliary $u$–dependence of the fields? (no dependence would make (A1) vanish). Performing the variation on (A1) I get:
\begin{equation}
\frac{3}{2}\int_0^1 du\, f_{abc}(\partial_u n^a)(\partial_t n^b)\delta n^c + \text{boundary term}, \nonumber
\end{equation}
which, however, does not straightforwardly integrate to the expression
\begin{equation}
\frac{1}{2}(\partial_t n^a) n^b \delta n^c. \nonumber
\end{equation}

Response: We thank the referee for pointing out this point, which enhances the completeness of our manuscript. The referee is right to point out that an additional assumption for $u$ should be explicitly stated. In the following, we present the detailed steps of the formal derivation of the generalized Landau-Lifshitz equation.

The generalized Wess-Zumino action we introduce is $$S_\mathrm{WZ} = \frac{1}{2}\int d^2 x dt \int_0^1 du f_{abc} n^a \partial_u n^b \partial_t n^c \,,$$
where (with a bit of abuse of notation) $n^a( x^\mu, u)$ is a smooth extension of the original field $n^a( x^\mu)$ such that $n^a( x^\mu, 0) = n^a_0$ (a constant) and $n^a( x^\mu, 1) = n^a( x^\mu)$. We also assume the condition for the variation, $\delta n^a (x^\mu, 0) = 0,\, \delta n^a (x^\mu, 1) = \delta n^a (x^\mu)$. Here $x^\mu$ are the 2+1 spacetime coordinates. Working out the variation of the action, we have
$$S_\mathrm{WZ}[n + \delta n] - S_\mathrm{WZ}[n] \approx \frac{1}{2}\int d^2 x dt \int_0^1 du f_{abc} \big[ \delta n^a \partial_u n^b \partial_t n^c + n^a \partial_u \delta n^b \partial_t n^c + n^a \partial_u n^b \partial_t \delta n^c \big] $$$$= \frac{1}{2}\int d^2 x dt \int_0^1 du f_{abc} \big[ \delta n^a \partial_u n^b \partial_t n^c + \partial_u \left( n^a \delta n^b \partial_t n^c \right) + \partial_t \left( n^a \partial_u n^b \delta n^c \right) - \partial_u n^a \delta n^b \partial_t n^c - n^a \delta n^b \partial_u \partial_t n^c - \partial_t n^a \partial_u n^b \delta n^c - n^a \partial_t \partial_u n^b \delta n^c \big]$$$$= \frac{1}{2}\int d^2 x dt \int_0^1 du f_{abc} \big[3\delta n^a \partial_u n^b \partial_t n^c + \partial_u \left( n^a \delta n^b \partial_t n^c \right) + \partial_t \left( n^a \partial_u n^b \delta n^c \right) \big]= \frac{1}{2}\int d^2 x dt \int_0^1 du f_{abc} \big[\partial_u \left( n^a \delta n^b \partial_t n^c \right) + \partial_t \left( n^a \partial_u n^b \delta n^c \right) \big]$$$$= \frac{1}{2}\int d^2 x dt \int_0^1 du f_{abc} \partial_u \left( n^a \delta n^b \partial_t n^c \right) = \frac{1}{2}\int d^2 x dt f_{abc} n^a \delta n^b \partial_t n^c \,.$$
Integration of the term $3\delta n^a \partial_u n^b \partial_t n^c$ vanishes due to the algebraic properties of $n^a$. Integration of the total derivative $ \partial_t \left( n^a \partial_u n^b \delta n^c \right)$ vanishes since $\delta n^a =0$ at the boundary in $t$. Integration over $u$ of $\partial_u \left( n^a \delta n^b \partial_t n^c \right)$ does not vanish and gives the final result. Therefore, we have
\begin{align}
\frac{\delta S_\mathrm{WZ}}{\delta n^a} = \frac{1}{2} f_{abc} \partial_t n^b n^c \,.
\end{align}

---

## Editorial Decision

in_refereeing